# M^3^ENet: A Multi-Modal Fusion Network for Efficient Micro-Expression Recognition

**DOI:** 10.3390/s25206276

**Published:** 2025-10-10

**Authors:** Ke Zhao, Xuanyu Liu, Guangqian Yang

**Affiliations:** 1Electronic Science and Technology Museum, University of Electronic Science and Technology of China, Chengdu 611731, China; 2School of Public Administration, University of Electronic Science and Technology of China, Chengdu 611731, China; 3State Key Laboratory of Photonics and Communications, Peking University, Beijing 100871, China; 4Department of Biomedical Engineering, Hong Kong Polytechnic University, Hong Kong 999077, China; guangqian.yang@connect.polyu.hk

**Keywords:** micro-expression recognition, multi-modal, convolution neural network, deep learning

## Abstract

Micro-expression recognition (MER) aims to detect brief and subtle facial movements that reveal suppressed emotions, discerning authentic emotional responses in scenarios such as visitor experience analysis in museum settings. However, it remains a highly challenging task due to the fleeting duration, low intensity, and limited availability of annotated data. Most existing approaches rely solely on either appearance or motion cues, thereby restricting their ability to capture expressive information fully. To overcome these limitations, we propose a lightweight multi-modal fusion network, termed M^3^ENet, which integrates both motion and appearance cues through early-stage feature fusion. Specifically, our model extracts horizontal, vertical, and strain-based optical flow between the onset and apex frames, alongside RGB images from the onset, apex, and offset frames. These inputs are processed by two modality-specific subnetworks, whose features are fused to exploit complementary information for robust classification. To improve generalization in low data regimes, we employ targeted data augmentation and adopt focal loss to mitigate class imbalance. Extensive experiments on five benchmark datasets, including CASME I, CASME II, CAS(ME)^2^, SAMM, and MMEW, demonstrate that M^3^ENet achieves state-of-the-art performance with high efficiency. Ablation studies and Grad-CAM visualizations further confirm the effectiveness and interpretability of the proposed architecture.

## 1. Introduction

Facial expression recognition (FER) plays a crucial role in understanding human emotions, with applications across various domains such as healthcare, education, surveillance, and biomedical engineering. By analyzing facial muscle movements, FER enables machines to detect emotional cues, facilitating natural human–computer interaction. In biomedical engineering, FER aids in monitoring patient emotions, detecting psychological conditions, and supporting therapeutic interventions for neurological disorders. Additionally, FER is increasingly used to analyze visitor experience in museums [1], providing insights into emotional engagement and satisfaction, which can help improve exhibits and overall visitor interaction. Among various facial expressions, micro-expressions (MEs) are particularly significant due to their involuntary, subtle, and fleeting nature, typically lasting between 1/25 and 1/5 s [2,3]. Unlike macro-expressions, MEs are automatic and difficult to control, making them reliable indicators of a person’s genuine emotional state. These brief facial movements often emerge when individuals attempt to suppress or conceal their true feelings, particularly in high-stakes or emotionally intense situations. As a result, MEs are crucial for applications in lie detection, criminal interrogation, and psychological assessments, where understanding concealed emotions is essential. In biomedical contexts, recognizing MEs holds great promise for monitoring the emotional and psychological health of patients, particularly those with conditions such as Parkinson’s disease, autism spectrum disorders, or facial paralysis. By detecting these subtle emotional cues, FER systems can provide real-time insights into a patient’s emotional well-being, supporting more effective therapies and interventions. Despite their potential, the recognition of MEs remains a challenging task due to their short duration, subtle intensity, and the limited availability of annotated datasets. To address these challenges, biomedical engineering approaches combining image processing, machine learning, and computer vision are being developed to create more efficient and accurate recognition systems, making MEs a valuable tool for both research and clinical applications.

Early research in micro-expression recognition predominantly relied on handcrafted features, which aimed to extract discriminative patterns from shallow visual cues. Among these, Local Binary Patterns on Three Orthogonal Planes (LBP-TOP) [4] and its variants have been widely adopted due to their compact representation, low computational cost, and robustness to illumination changes. Several extensions have been proposed to enhance LBP’s spatiotemporal modeling ability, such as Local Binary Pattern with Six Intersection Points (LBP-SIP) [5], Local Binary Pattern with Mean Orthogonal Planes (LBP-MOP) [6], and Spatiotemporal Completed Local Quantization Patterns (STCLQP) [7]. These methods extract local texture descriptors across temporal frames to represent subtle facial movements. In parallel, optical flow-based techniques have also been explored to capture the subtle motions between consecutive frames. Optical flow estimates pixel-wise movement by computing brightness changes, making it suitable for detecting subtle facial muscle dynamics. Representative approaches include Optical Strain Features (OSFs) [8], Optical Strain Weights (OSWs) [9], Fuzzy Histogram of Oriented Optical Flow (FHOOF) [10], Main Directional Mean Optical Flow (MDMO) [11], and the more recent Bi-Weighted Oriented Optical Flow (Bi-WOOF) [12]. While these handcrafted approaches offer interpretability and require relatively fewer training samples, they often suffer from limited discriminative power and generalization capability, especially under uncontrolled conditions involving variations in head pose, illumination, and individual differences. Moreover, the reliance on carefully designed descriptors and prior knowledge limits their scalability and adaptability in practical applications.

With the advent of deep learning, particularly convolutional neural networks (CNNs), researchers have increasingly explored data-driven methods for micro-expression recognition. Early attempts, such as the work by Patel et al. [13], adopted standard CNN architectures but failed to outperform traditional handcrafted descriptors due to overfitting and insufficient data. Subsequent works addressed these limitations by integrating domain-specific priors. For instance, Peng et al. [14] introduced a dual-temporal-scale CNN using optical flow inputs to capture multi-scale motion features.

To further exploit temporal dynamics, some methods combined CNNs with recurrent structures such as LSTM [15] or introduced frame alignment techniques (e.g., TIM [16]) to standardize input lengths. Liong et al. [17] proposed a Shallow Triple-Stream 3D CNN (STSTNet) to process horizontal, vertical, and optical strain components of apex-frame optical flow, achieving promising performance with low model complexity. More recently, inspired by the success of Transformers in modeling long-range dependencies, Wang et al. [18] adopted a Transformer encoder to extract global spatial representations from long-term optical flow sequences, further combined with LSTM for temporal modeling.

Despite these advances, most existing deep learning methods remain confined to a single modality, typically either RGB sequences or optical flow. As a result, they fail to exploit the complementary nature of appearanceand motion information. Empirical studies have shown that while optical flow captures subtle motion cues, RGB frames retain rich appearance and texture details, both of which are crucial for accurately identifying fleeting facial expressions. Relying solely on one modality often limits the representational power of the network. It leads to suboptimal performance, especially under challenging scenarios such as low intensity, ambiguous expressions, or inter-subject variation.

These limitations highlight the necessity of multi-modal fusion, which aims to integrate appearance and motion features in a unified framework. By jointly modeling RGB and optical flow representations, a well-designed multi-modal system can effectively leverage their complementary strengths, improve robustness to visual variability, and enhance overall recognition accuracy. This motivates us to develop a lightweight, multi-stream architecture that fuses features at an early stage, enabling more effective and efficient micro-expression recognition.

To address these limitations, we propose a novel five-stream CNN framework for unified feature extraction from both RGB frames and optical flow data. Specifically, we extract three types of optical flow components—horizontal, vertical, and optical strain—between the onset and apex frames to capture subtle motion dynamics. In parallel, we use three key RGB frames—the onset, apex, and offset frames—as inputs to extract appearance-based features. This design not only reduces the computational burden compared to processing full video clips but also eliminates redundant frames while preserving crucial temporal dynamics of micro-expressions. We employ two independent subnetworks to learn modality-specific representations for optical flow and RGB inputs, respectively. These features are subsequently fused at the feature level to leverage their complementary information. The final classification is achieved through a fully connected layer followed by a softmax function. Furthermore, to mitigate the limited data availability in current micro-expression datasets, we design a comprehensive data augmentation strategy, including image flipping, rotation, and additive noise, which significantly boosts performance and enhances generalization across different datasets. Our contributions are summarized below:We propose a unified five-stream CNN architecture that effectively integrates both optical flow and RGB modalities, enabling joint modeling of motion and appearance cues critical for micro-expression recognition.To tackle sample scarcity and label imbalance in micro-expression datasets, we introduce a customized data augmentation pipeline and incorporate focal loss, leading to consistent performance gains across standard benchmarks.We conduct extensive experiments on publicly available datasets and demonstrate that our approach achieves state-of-the-art (SOTA) performance while maintaining high computational efficiency, offering a strong baseline for robust and interpretable multi-modal micro-expression analysis.

## 2. Related Work

### 2.1. ME Datasets

The construction of high-quality ME datasets is crucial for accurate ME recognition. Unlike other computer vision tasks’ datasets, such as ImageNet [19] or CIFAR-10 [20], where data collection is relatively straightforward, MEs are characterized by their spontaneous, involuntary, and fleeting nature, often triggered under high-stakes or emotionally intense conditions. This makes both data acquisition and annotation significantly more challenging.

In recent years, several spontaneous ME datasets have been introduced, including SMIC [21] and its extended version SMIC-E, CASME [22], CASME II [23], CAS(ME)^2^ [24], SAMM [25], and the Macro- and Micro-Expression Warehouse (MMEW) [26]. This paper focuses on these spontaneous datasets.

The following summarizes the key characteristics of the most widely used spontaneous ME datasets, highlighting their data collection protocols, annotation strategies, and contributions to the field. These datasets provide the foundation for training and evaluating ME recognition models under various settings.

**SMIC [21]:** SMIC (Spontaneous Micro-Expression Corpus) was the first public spontaneous ME dataset, containing 71 video clips from 20 participants. It includes three modalities: high speed (HS, 100 fps), visible light (VIS, 25 fps), and near infrared (NIR), captured while participants watched emotion-inducing videos under instructions to maintain a neutral expression. Each clip is labeled via participant self-report as “positive,” “negative,” or “surprise,” and FACS-trained coders provide frame-level annotations to ensure reliability.**CASME [22]:** CASME (Chinese Academy of Sciences Micro-Expression) contains 195 spontaneous MEs elicited from 35 participants. Videos were captured at 60 fps in a controlled setting, with coding of onset, apex, and offset frames. Emotion labels stem from a combination of participant self-reports and AU-based FACS.**CASME II [23]:** CASME II builds on CASME with 247 spontaneous micro-expression samples from 26 participants. Videos were recorded at an improved 200 fps and 280 × 340 px resolution. Labeling combines AUs, subjective reports, and contextual video information, with ambiguous cases tagged as “others”. It is the most widely used dataset for MER.**CAS(ME)^2^ [24]:** CAS(ME)^2^ extends CASME II by capturing both spontaneous micro- and macro-expressions from the same 22 participants. It contains 57 micro-expression samples and 300 cropped macro-expression samples, all recorded at 30 fps. In addition, 87 long video sequences are provided for joint evaluation of expression spotting and recognition. Emotion labels are assigned based on a combination of facial action units (AUs), the nature of the emotion-inducing videos, and participant self-reports, categorized into “positive,” “negative,” “surprise,” and “others.” The dataset facilitates research into cross-scale expression dynamics and supports both recognition and temporal spotting tasks.**SAMM [25]:** SAMM (Spontaneous Actions and Micro-Movements) is a high-resolution spontaneous ME dataset comprising 159 micro-expression samples recorded at 200 or 300 fps under controlled lighting conditions, with facial regions captured at 2040 × 1088 resolution. The dataset emphasizes diversity by including 32 participants from 13 different ethnicities. Each ME is annotated with onset, apex, and offset frames, as well as corresponding facial action units (AUs). Emotion categories include contempt, disgust, fear, anger, sadness, happiness, and surprise. To induce genuine emotional responses, participant-specific video stimuli and incentive-based protocols were employed. An extended version, SAMM Long Videos [27], provides 147 long videos containing both micro- and macro-expressions, further supporting research in expression spotting across temporal scales.**MMEW [26]:** MMEW (Macro- and Micro-Expression Warehouse) is a multi-modal database containing 300 micro- and 900 macro-expression clips, covering six basic emotions (happiness, surprise, anger, disgust, fear, sadness). Data are available in RGB, depth, and infrared formats, with naming conventions that simplify emotion-specific retrieval. It supports deep learning research due to its scale and richness.**Composite Dataset [28] and CMED [29]:** The Composite Dataset (from MEGC2019) merges CASME II, SAMM, and SMIC-HS, unifying emotion labels into positive, negative, and surprise to standardize evaluation. The Compound Micro-Expression Dataset (CMED) aggregates MEs from CASME, CASME II, CAS(ME)^2^, SMIC-HS, and SAMM, categorizing expressions into basic and compound emotions, reflecting the psychological realism of naturally co-occurring affective states.

Table 1 provides a detailed comparison of existing spontaneous ME datasets. Despite their contributions, these corpora (e.g., SMIC, CASME, CASME II, CAS(ME)2, SAMM, and MMEW) still face critical limitations that hinder the external validity and scalability of MER systems. First, sample sizes are small and highly imbalanced: CAS(ME)^2^ contains only 57 clips, CASME II has 247, SAMM 159, and even MMEW provides just 300 micro-expression clips, with minority categories such as fear or contempt especially underrepresented. This scarcity often forces studies to focus only on categories with more than ten samples, leading to coarse-grained recognition and overfitting. Second, ecological validity is limited because most recordings were collected under controlled laboratory conditions with uniform illumination and frontal poses, far from in-the-wild scenarios with head movements, occlusions, or compression artifacts. Third, population diversity remains narrow: apart from SAMM’s effort to include multiple ethnicities, demographic coverage is restricted, and heterogeneous frame rates, resolutions, and label taxonomies complicate cross-dataset learning and fair comparison.

Looking forward, several directions are vital for the community: (i) building larger, demographically diverse, and ecologically valid corpora across naturalistic contexts (e.g., social interaction, telehealth, surveillance); (ii) standardizing annotation protocols by harmonizing AU coding, emotion taxonomies, and onset–apex–offset timestamps, while establishing cross-dataset benchmarks to assess domain shift; and (iii) leveraging synthetic augmentation (e.g., GANs, diffusion models) to mitigate class imbalance without distorting micro-expression dynamics.

### 2.2. Deep Learning-Based Micro-Expression Recognition Pipeline

Deep learning has significantly advanced micro-expression recognition (MER) by enabling models to automatically extract discriminative features from raw visual data. A typical deep MER pipeline consists of three key stages: advanced preprocessing, specialized network architectures, and training strategies tailored to the challenges of MER.

**Preprocessing.** The pipeline typically begins with face detection and alignment, using either traditional methods (e.g., Viola-Jones [30]) or modern CNN-based detectors [31]. To amplify the subtle facial movements of micro-expressions, motion magnification techniques such as Eulerian Video Magnification [32] are widely applied. Temporal normalization ensures that all video clips are resampled to a consistent length, often via the Temporal Interpolation Model (TIM) or CNN-based frame interpolation [33]. To focus on informative facial areas, researchers frequently extract task-relevant regions of interest (ROIs), such as those defined by the Facial Action Coding System (FACS) or anatomical priors [34]. Given the limited size of existing MER datasets, data augmentation is essential, extending from basic techniques (e.g., flipping, cropping) to more sophisticated strategies such as multi-scale motion magnification or GAN-based data synthesis [35,36].

**Network Architectures.** Early MER models relied on shallow 2D CNNs applied to key frames (e.g., apex frames) to reduce overfitting. More recent works adopt deeper architectures with pretrained backbones (e.g., VGG, ResNet), and incorporate spatiotemporal modeling via 3D CNNs or recurrent layers like LSTM and GRU [37]. Many designs adopt CNN-RNN cascades to model frame-wise evolution. Beyond these, Graph Convolutional Networks (GCNs) have been introduced to capture spatial or temporal dependencies among facial landmarks or action units [38]. Other variants incorporate capsule networks or attention modules (e.g., CBAM) to enhance sensitivity to subtle spatial changes. Transformer-based models, such as HTNet [18], divide the face into semantically meaningful regions (e.g., eyes, mouth) and use hierarchical self-attention to learn both local dynamics and global interactions. Overall, MER models span single-stream 2D/3D CNNs, CNN-RNN hybrids, multi-stream fusion networks, GCNs, and Transformers.

**Training Strategies.** Most MER models employ the standard softmax (cross-entropy) loss for classification, but often combine it with auxiliary objectives to address the high intra-class similarity and severe class imbalance in MER datasets. Metric learning-based losses, such as triplet loss [39] and center loss, are used to improve feature space discriminability. Focal loss [37] is frequently adopted to emphasize hard or under-represented samples. These combined objectives have been shown to improve model robustness and generalization under limited supervision.

In summary, recent deep learning-based MER systems follow a consistent yet evolving pipeline: comprehensive preprocessing to enhance subtle motions and temporal consistency, specialized architectures for fine-grained spatiotemporal representation, and training strategies designed to cope with limited and imbalanced data. These innovations collectively address the fundamental challenges of micro-expression recognition, including low motion intensity, short duration, and scarce annotations.

### 2.3. Multi-Modal Approaches for Micro-Expression Recognition

To better exploit the subtle and brief nature of micro-expressions (MEs), recent studies have explored multi-modal inputs that capture complementary aspects of facial dynamics. Instead of relying solely on RGB video frames, many works integrate various visual cues such as apex frames, optical flow, and facial landmarks to enhance representation quality. For example, Liu et al. [40] utilize apex frames to capture peak expression details while leveraging optical flow to model temporal motion, effectively extracting both spatial and temporal features. Expanding on this idea, Song et al. [41] introduce local facial regions from the apex frame to model the relationships among facial sub-areas, improving robustness to expression variability. Sun et al. [42] combine optical flow and full video sequences to fully mine temporal dynamics. Furthermore, inspired by the success of facial landmarks in expression analysis, Kumar et al. [43] propose fusing landmark graphs with optical flow to enhance the discriminative capacity of the network.

These multi-modal designs have led to state-of-the-art performance by maximizing the use of limited ME data. However, several limitations remain. First, the input modalities used by existing methods often fall into two extremes: either full video sequences, which are computationally expensive and may contain redundant or irrelevant frames, or single RGB frames (typically the apex frame), which may overlook important temporal cues and fail to fully exploit the visual modality, leading to suboptimal performance. Second, despite their empirical success, most current networks lack interpretability. The fusion of heterogeneous inputs is typically implemented through black-box neural modules, offering little insight into which features or modalities contribute most to the final prediction. This limits the transparency and trustworthiness of such systems, especially in sensitive applications like deception detection or clinical diagnosis.

Overall, while multi-modal approaches effectively enhance MER performance by integrating complementary visual cues, there remains a pressing need for more efficient input selection strategies and interpretability analysis that can offer both high accuracy and transparency.

## 3. Methods

Our method is designed to accurately recognize micro-expressions by leveraging both appearance and motion cues. As illustrated in Figure 1, the overall framework comprises four key components: a preprocessing module, RGB and optical flow feature encoders, and an output module. Each module collaboratively extracts and fuses multi-modal information.

### 3.1. Preprocessing Module

The preprocessing module is designed to prepare the input data for feature extraction, comprising four essential steps: face detection, alignment, apex frame spotting, and optical flow computation.

We adopt the OpenCV library with the Local Binary Features (LBF) model to perform face detection and landmark tracking across video frames. Subsequently, all frames are resized to H×W pixels, resulting in standardized inputs represented as $I_{i}\in R^{3\times H\times W}$.

To reduce computational redundancy while preserving the most informative motion cues, we perform apex frame spotting. Specifically, for each video clip, domain-expert annotators label three key frames: the onset frame (beginning of the expression), the apex frame (moment of peak expression), and the offset frame (end of the expression). These frames are respectively denoted as $I_{\mathrm{onset}}$, $I_{\mathrm{apex}}$, and $I_{\mathrm{offset}}$. This sparse frame selection strategy significantly reduces data volume and computational cost while retaining the critical temporal information necessary for micro-expression analysis. The selected RGB frames $\left\{I_{\mathrm{onset}},I_{\mathrm{apex}},I_{\mathrm{offset}}\right\}$ are subsequently fed into the RGB feature extraction module to encode static spatial features corresponding to different temporal stages of expression development.

In parallel, to capture the subtle dynamic changes between frames, we compute optical flow between the onset and apex frames using the Dual TV-L1 algorithm [44], which is known for its robustness to illumination variation and preservation of motion discontinuities. This yields a dense motion field *O* that encodes pixel-wise displacements in both horizontal and vertical directions:(1)O={(u(x,y),v(x,y))∣x=1,2,...,H;y=1,2,...,W}
where u(x,y) and v(x,y) represent the horizontal and vertical motion vectors at pixel (x,y), respectively.

To further characterize localized facial muscle deformation, we derive the optical strain map ϵ(x,y) from the flow field. This is achieved by computing the spatial gradients of *u* and *v*, forming the strain tensor:(2)$$\epsilon =\left[\begin{array}{cc} \epsilon_{xx}=\frac{\partial u}{\partial x} & \epsilon_{xy}=\frac{1}{2}\left(\frac{\partial u}{\partial y}+\frac{\partial v}{\partial x}\right)\\ \epsilon_{yx}=\frac{1}{2}\left(\frac{\partial v}{\partial x}+\frac{\partial u}{\partial y}\right) & \epsilon_{yy}=\frac{\partial v}{\partial y} \end{array}\right]$$
where $\epsilon_{xx}$ and $\epsilon_{yy}$ denote normal strain, and $\epsilon_{xy}$ and $\epsilon_{yx}$ represent shear strain components.

The overall magnitude of strain at each pixel is quantified using the Frobenius norm of the strain tensor:(3)$$\epsilon \left(x,y\right)=\sqrt{\epsilon_{xx}^{2}+\epsilon_{yy}^{2}+\epsilon_{xy}^{2}+\epsilon_{yx}^{2}}.$$

Finally, the computed optical flow maps are organized as three separate single-channel feature maps: the horizontal component $O_{u}=u(x,y)\in R^{1\times H\times W}$, the vertical component $O_{v}=v(x,y)\in R^{1\times H\times W}$, and the optical strain map $O_{\epsilon }=\epsilon (x,y)\in R^{1\times H\times W}$. These are fed individually into the optical flow feature extraction branch for spatiotemporal representation learning.

### 3.2. Multi-Modal Feature Extraction

The multi-modal feature extraction stage is designed to learn complementary representations from static RGB appearance and dynamic motion features. To this end, we construct two separate encoding branches: one for RGB frames and one for optical flow maps.

#### 3.2.1. RGB Feature Encoder

To encode the appearance information of facial expressions at different time points, we extract features from the onset, apex, and offset frames using a shared shallow convolutional network. Each frame $I\in R^{3\times H\times W}$ is passed through the following convolutional block:(4)$$F^{\prime }=\mathrm{ReLU}\left(\mathrm{BN}\left({\mathrm{Conv}}_{7\times 7}\left(I\right)\right)\right).$$(5)$${\mathrm{Conv}}_{\mathrm{RGB}}\left(I\right)=\mathrm{ReLU}\left(\mathrm{BN}\left({\mathrm{Conv}}_{3\times 3}\left(F^{\prime }\right)\right)\right).$$

This results in three intermediate feature maps: $F_{\mathrm{onset}},F_{\mathrm{apex}},F_{\mathrm{offset}}\in R^{D_{rgb}\times H\times W}$, where $D_{rgb}$ denotes the feature dimension.

To enhance the model’s ability to capture temporal contrast and subtle visual changes in facial appearance, we perform feature concatenation between the apex frame and its adjacent frames:(6)$$F_{\mathrm{rgb},1}=\mathrm{Concat}(F_{\mathrm{onset}},F_{\mathrm{apex}}).$$(7)$$F_{\mathrm{rgb},2}=\mathrm{Concat}(F_{\mathrm{offset}},F_{\mathrm{apex}}).$$

This design allows the network to explicitly model the relative differences in appearance between the apex frame and both the start and end of the expression. Compared to directly processing each frame individually, this strategy emphasizes the local variations that are critical for distinguishing subtle micro-expressions.

Each concatenated feature map is then passed through an independent residual block:(8)$$F_{\mathrm{rgb},i}^{\prime }={\mathrm{ResBlock}}_{\mathrm{RGB}}\left(F_{\mathrm{rgb},i}\right),\;i=1,2$$

Finally, the outputs are flattened into two 1D feature vectors:(9)$$f_{\mathrm{rgb},i}=\mathrm{Flatten}\left(F_{\mathrm{rgb},i}^{\prime }\right)\in R^{d},\;i=1,2$$

These two vectors jointly represent the RGB-based spatial appearance features for subsequent fusion and classification.

#### 3.2.2. Optical Flow Feature Encoder

The optical flow encoder processes the horizontal flow map $O_{u}$, vertical flow map $O_{v}$, and optical strain map $O_{\epsilon }$, each treated as a single-channel input. For each flow map $O\in R^{1\times H\times W}$, we first apply a convolutional layer to project it into a high-dimensional feature space:(10)$$F_{O}^{\left(0\right)}=\mathrm{ReLU}\left(\mathrm{BN}\left({\mathrm{Conv}}_{3\times 3}\left(O\right)\right)\right)\in R^{D_{of}\times H\times W}.$$

Then, *N* stacked residual blocks are applied to progressively extract deeper motion representations:(11)$$F_{O}^{\left(N\right)}={\mathrm{ResBlock}}^{\left(N\right)}\left(\cdots {\mathrm{ResBlock}}^{\left(1\right)}\left(F_{O}^{\left(0\right)}\right)\right).$$

Finally, each output feature map is flattened into a 1D vector:(12)$$f_{O}=\mathrm{Flatten}\left(F_{O}^{\left(N\right)}\right)\in R^{d}.$$
where $O\in \{O_{u},O_{v},O_{\epsilon }\}$, and the resulting vectors $f_{O_{u}},f_{O_{v}},f_{O_{\epsilon }}$ form the optical flow-based motion representation.

### 3.3. Output Module

The output module integrates the appearance and motion representations for final classification. Specifically, the two RGB-based feature vectors $f_{\mathrm{rgb},1}$ and $f_{\mathrm{rgb},2}$, and the three flow-based vectors $f_{O_{u}},f_{O_{v}},f_{O_{\epsilon }}$ are concatenated into a unified representation:(13)$$f_{\mathrm{concat}}=\mathrm{Concat}\left(f_{\mathrm{rgb},1},f_{\mathrm{rgb},2},f_{O_{u}},f_{O_{v}},f_{O_{\epsilon }}\right).$$

The fused feature vector $f_{\mathrm{concat}}\in R^{5d}$ is then passed through two fully connected layers with ReLU activation:(14)$$h_{1}=\mathrm{ReLU}({\mathrm{FC}}_{1}\left(f_{\mathrm{concat}}\right)).$$(15)$$h_{2}=\mathrm{ReLU}({\mathrm{FC}}_{2}\left(h_{1}\right)).$$

Finally, the classification logits are computed and normalized using the softmax function to obtain the predicted probability distribution:(16)$$\hat{y}=\mathrm{Softmax}\left({\mathrm{FC}}_{\mathrm{out}}\left(h_{2}\right)\right)\in R^{C_{d}}.$$

Here, $C_{d}$ denotes the number of emotion categories in dataset *d*.

## 4. Experiments and Results

In this section, we first describe the experimental settings, including the used datasets, evaluation metrics, baseline methods for comparison, and implementation details. We then evaluate the performance of the proposed M^3^ENet method across multiple ME datasets, followed by interpretability analysis and ablation studies to validate the reliability and effectiveness of the proposed framework.

### 4.1. Experimental Setup

#### 4.1.1. Datasets

To demonstrate the effectiveness of the proposed framework, we conduct extensive MER experiments on five widely used public datasets: CASME, CASME II, SAMM, CAS(ME)^2^, and MMEW. Detailed descriptions of these datasets are provided in Section 2. An illustration of the data augmentation strategies used in our framework is shown in Figure 2.

To address the issue of limited training data, we apply data augmentation techniques, including image flipping, rotation, and additive noise. Each original sequence is augmented into five variants, effectively increasing the dataset size fivefold. These augmentations enhance data diversity, mitigate overfitting, and improve generalization. Representative examples of the augmented data are shown to illustrate the effectiveness of these transformations, as illustrated in Figure 3.

For all five datasets, we adopt the leave-one-subject-out (LOSO) evaluation protocol to ensure a fair and reliable assessment. Specifically, in each fold, the samples of one subject are used as the test set, while the remaining samples are used for training. This protocol prevents identity overlap between training and testing sets, thus avoiding subject-dependent bias and enhancing the reliability of evaluation results.

#### 4.1.2. Evaluation Metrics

To comprehensively evaluate the performance of MER models, we report the following metrics: precision, recall, F1-score, accuracy, and the area under the ROC curve (AUC). Their definitions are given below:(17)$$\begin{array}{c} \mathrm{Precision}=\frac{TP}{TP+FP}, \end{array}$$(18)$$\begin{array}{c} \mathrm{Recall}=\frac{TP}{TP+FN}, \end{array}$$(19)$$\begin{array}{c} F1\text{-}\mathrm{score}=\frac{2\cdot \mathrm{Precision}\cdot \mathrm{Recall}}{\mathrm{Precision}+\mathrm{Recall}}, \end{array}$$(20)$$\begin{array}{c} \mathrm{Accuracy}=\frac{TP+TN}{TP+TN+FP+FN}, \end{array}$$(21)$$\begin{array}{c} \mathrm{AUC}=\sum_{i=1}^{N-1}\left(FPR_{i+1}-FPR_{i}\right)\cdot \frac{TPR_{i+1}+TPR_{i}}{2}, \end{array}$$
where TP, TN, FP, and FN denote the numbers of true positives, true negatives, false positives, and false negatives, respectively. In the AUC formula, $TPR_{i}=\frac{TP_{i}}{TP_{i}+FN_{i}}$ and $FPR_{i}=\frac{FP_{i}}{FP_{i}+TN_{i}}$ represent the true positive rate and false positive rate at the *i*-th threshold, and *N* denotes the number of evaluation thresholds.

#### 4.1.3. Baselines

To validate the effectiveness of the proposed method, we conduct comprehensive comparisons with a diverse set of baseline approaches, including generic deep learning models, ME-specific single-modal methods, and ME-specific multi-modal methods.

**Generic Deep Learning Methods**: These methods were originally designed for general image recognition tasks and are widely used as backbones or baselines in MER studies. Despite not being tailored for micro-expression recognition, they offer a strong benchmark for evaluating feature extraction capabilities.

**AlexNet** [45]: A classic CNN architecture consisting of five convolutional layers and three fully connected layers. It is used here as a shallow baseline for visual feature learning.**VGG-16** [46]: A deeper architecture with small convolution kernels (3 × 3) and a uniform design. VGG-16 is known for its strong representation ability and is commonly used for fine-grained recognition tasks.**ResNet-18** [47]: Incorporates residual connections to mitigate the vanishing gradient problem in deep networks. Its lightweight version (ResNet-18) is suitable for small-scale datasets like MER.**GoogLeNet (Inception v1)** [48]: Utilizes multi-scale convolution within inception modules, allowing more expressive features with fewer parameters.

**Single-Modal ME-Specific Methods**: These approaches are explicitly designed for MER and typically leverage temporal modeling techniques (e.g., LSTM or 3D CNNs) or incorporate domain-specific priors to better capture the subtle and transient nature of micro-expressions. They usually focus on a single modality, such as RGB or optical flow.

**Off-ApexNet** [49]: A dual-stream CNN model that captures subtle facial motion by computing optical flow between the onset and apex frames. The network separately processes the horizontal and vertical components of flow to learn discriminative motion features. It is one of the earliest deep learning frameworks tailored for MER tasks.**STSTNet** [17]: A lightweight three-stream 3D CNN that processes horizontal flow, vertical flow, and optical strain simultaneously. By employing shallow temporal convolution, it effectively models the short-term dynamics of micro-expressions. The architecture is highly efficient and well suited for small-scale MER datasets.**HTNet** [18]: A hierarchical Transformer designed for micro-expression recognition. It divides the face into four regions and uses local self-attention to capture subtle muscle movements, while an aggregation layer models interactions between eye and lip areas.**LAENet** [50]: A local self-attention encoding network that focuses on critical facial regions with subtle muscle movements. It incorporates spatial attention to selectively enhance local features relevant to micro-expression.**SSRLTS-ViT** [51]: SSRLTS-ViT is a three-stream Vision Transformer baseline for micro-expression recognition that learns from three optical flow components to capture subtle facial motion with global context.

**Multi-Modal ME-Specific Methods**: These methods extend single-modal approaches by combining multiple modalities (e.g., RGB, optical flow) to improve discrimination. They often use fusion strategies or cross-modal learning to exploit inter-modal relationships.

**CNNCapsNet** [40]: This method integrates five input streams, including vertical and horizontal optical flow between onset–apex and apex–offset, as well as the apex grayscale image. A multi-stream CNN is used for feature extraction, followed by a Capsule Network for final classification.

#### 4.1.4. Implementation Details

The experiments were conducted on an Ubuntu 20.10 system equipped with 4 Intel Xeon Platinum 8375C CPUs (Intel Corporation, Santa Clara, CA, USA), a single NVIDIA GeForce RTX 4090 GPU (NVIDIA Corporation, Santa Clara, CA, USA), and 256 GB of RAM. The model was trained for 50 epochs with a batch size of 128 using the AdamW optimizer, a cosine annealing learning rate schedule, and a weight decay of 0.05. The initial learning rate was set to 0.001, and the momentum coefficient β was fixed at 0.999. The input images were uniformly resized to 48×48.

To address the issue of class imbalance, where certain expression categories (e.g., “disgust”) are heavily underrepresented, we employ the focal loss [52], which dynamically adjusts the contribution of each sample to the loss based on its classification difficulty.

The focal loss is formulated as(22)$$L_{FL}=-\alpha_{t}{\left(1-p_{t}\right)}^{\gamma }\log\left(p_{t}\right)$$
where $p_{t}$ is the model’s predicted probability for the true class label, $\alpha_{t}\in \left(0,1\right)$ is a class-specific weighting factor to handle imbalance, and γ≥0 is the focusing parameter that down-weights well-classified examples (i.e., those with high $p_{t}$) and focuses learning on hard, misclassified samples.

When γ=0, the focal loss reduces to standard cross-entropy. As γ increases, the relative loss assigned to easier examples decreases, thus encouraging the model to pay more attention to underrepresented or ambiguous samples. This property makes focal loss particularly suitable for micro-expression datasets where some emotion categories may be sparsely distributed.

In our experiments, we set γ=2 to effectively focus learning on hard samples. The class-wise weighting factor $\alpha_{t}$ is computed based on the inverse frequency of each category as(23)$$\alpha_{t}=1-\frac{n_{t}}{N},$$
where $n_{t}$ denotes the number of samples in class *t*, ensuring higher weights for underrepresented categories.

### 4.2. Performance Evaluation

#### 4.2.1. Overall Performance

Table 2 demonstrates that M^3^ENet consistently outperforms all baseline methods across all five datasets, achieving state-of-the-art (SOTA) performance. Specifically, M^3^ENet attains an average accuracy of 81.5%, an F1-score of 0.736, and an AUC of 0.953. These results highlight the superiority of our network architecture in effectively extracting and integrating features from both RGB and optical flow modalities, leading to significant improvements over single-modal methods. In contrast, existing multi-modal approaches often suffer from suboptimal modality selection and inadequate fusion strategies, resulting in limited performance gains. By carefully designing and aligning complementary representations across modalities, M^3^ENet enables more accurate and robust micro-expression recognition. Additionally, the confusion matrices of M^3^ENet on all five datasets (Figure 4) provide detailed insights into the model’s performance across different micro-expression categories.

#### 4.2.2. Interpretability Analysis

This interpretability is of paramount importance for enhancing the trustworthiness and user acceptance of the model, particularly in applications requiring high precision and reliability, such as mental health assessments and interpersonal interaction analysis. In our study, we employed Grad-Cam [53] visualization techniques to elucidate the decision-making processes of our micro-expression recognition network. The visualizations depicted in Figure 5 demonstrate the model’s capability to accurately identify regions of the face that are crucial for micro-expression recognition, often corresponding to action units (AUs). The Grad-Cam visualizations not only validate the model’s precision in recognizing micro-expressions but also provide transparency into the model’s decision-making process, thereby enhancing its interpretability.

#### 4.2.3. Ablation Analysis

To validate the effectiveness of the proposed approach, we conduct ablation studies by altering the network architecture and training strategies. Specifically, the training strategy variant involves removing data augmentation (w/o Data augmentation), while architectural variants include removing the RGB input stream (w/o RGB input), removing the optical flow input stream (w/o Optical flow input), using only a single RGB image as input (Single RGB input), and using the shuffled RGB image as input (Shuffled RGB input).

The ablation results in Table 3 clearly demonstrate the importance of each component in our framework. First, removing data augmentation (w/o Data augmentation) leads to a dramatic drop in all evaluation metrics, confirming the critical role of data diversity in preventing overfitting and enhancing generalization. Second, when comparing the two input modalities, removing the optical flow stream (w/o Optical flow input) causes a more severe performance degradation than removing the RGB stream (w/o RGB input). This highlights the indispensable contribution of motion information in capturing subtle facial dynamics for micro-expression recognition. Furthermore, the results of the Single RGB input variant indicate that a single static frame provides only limited discriminative cues, while the poor performance with Shuffled RGB input demonstrates that it is the texture information within RGB frames, rather than the RGB modality itself, that effectively guides the model’s decisions together, these findings validate that both spatial and temporal information, along with effective training strategies, are indispensable for the superior performance of our proposed M^3^ENet.

#### 4.2.4. Efficiency and Complexity Analysis

By extracting key frames from the video stream, we substantially reduce the dimensionality of the network input, which simplifies deployment in practical systems. To quantify computational budget, we report three standard metrics: floating-point operations (FLOPs), parameter count, and per-frame inference latency. As summarized in Table 4, Transformer-based baselines (e.g., SSRLTS-ViT and HTNet) exhibit prohibitive latency and computational cost, which undermines real-time deployment. Conventional unimodal models are lighter, yet their accuracy is often insufficient for production use. Our method, M^3^ENet, attains a more favorable accuracy–efficiency trade-off: by leveraging multi-modal feature fusion, it markedly improves performance while maintaining a lightweight computational footprint, facilitating deployment in resource-constrained, real-time settings.

## 5. Conclusions

In this paper, we propose a novel network architecture for micro-expression recognition using multi-modal inputs, which is applicable to a variety of real-world scenarios such as visitor experience analysis in museum environments. We demonstrate its superior performance on five publicly available datasets (CASME I, CASME II, CAS(ME)^2^, SAMM, and MMEW). Specifically, the network takes optical flow data and RGB images from the onset, apex, and offset frames as inputs, with dedicated encoders designed for each modality to extract complementary features. Data augmentation and focal loss are employed to address data scarcity and class imbalance, significantly improving recognition accuracy across datasets. Experimental results show that, thanks to the multi-modal input design and the carefully crafted architecture, M^3^ENet significantly outperforms both conventional general-purpose models and existing single-modal and multi-modal MER methods. Furthermore, we employ Grad-CAM to visualize the model’s attention, demonstrating the interpretability and rationality of its decision-making. Comprehensive ablation studies further validate the importance of each proposed component to the overall model performance.

## Figures and Tables

**Figure 1 sensors-25-06276-f001:**
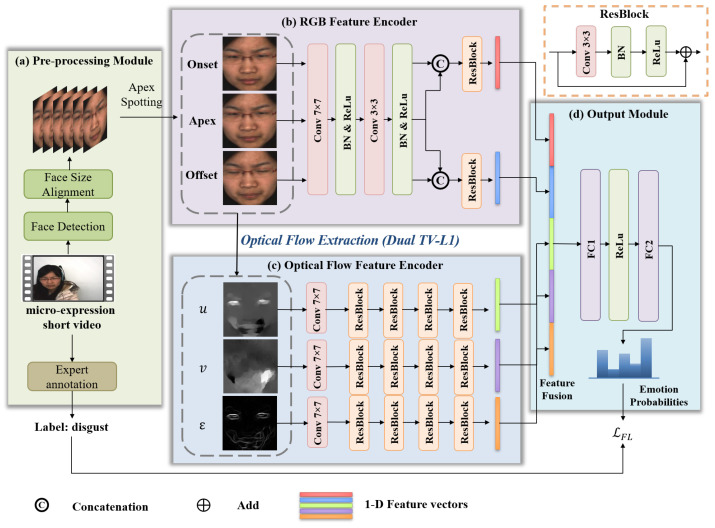
Illustration of the proposed five-stream architecture for micro-expression analysis. The network consists of a preprocessing module, an RGB feature encoder, an optical flow feature encoder, and an output module.

**Figure 2 sensors-25-06276-f002:**
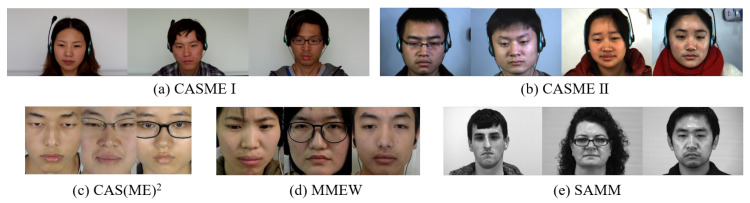
Illustration of representative micro-expression samples from multiple datasets used in this work.

**Figure 3 sensors-25-06276-f003:**

Illustration of the proposed data augmentation strategy with random rotation angles and noise levels across samples.

**Figure 4 sensors-25-06276-f004:**
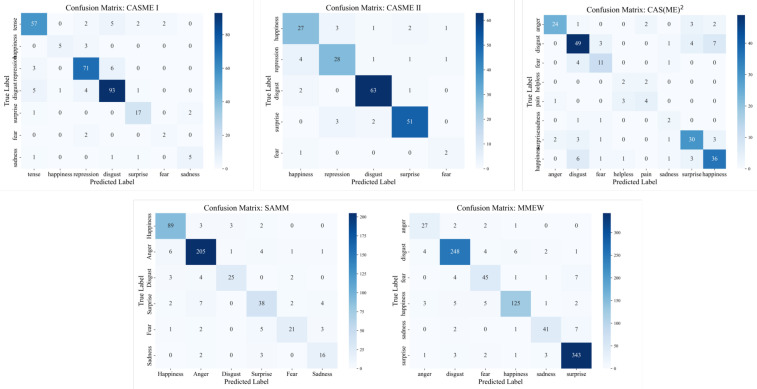
Confusion matrices of model predictions on the five ME datasets.

**Figure 5 sensors-25-06276-f005:**
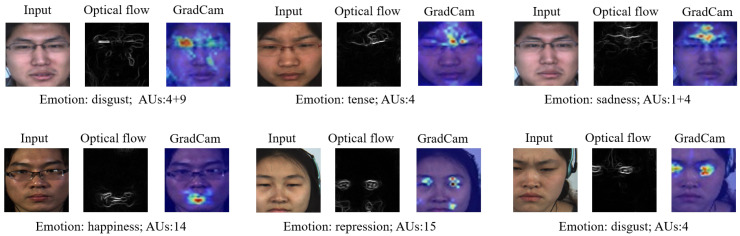
Illustration of Grad-CAM visualizations highlighting model attention on key facial regions across micro-expression types and datasets.

**Table 1 sensors-25-06276-t001:** Summary of previous spontaneous ME datasets.

Dataset	SMIC [21]	CASME [22]	CASME II [23]	CAS(ME)^2^ [24]	SAMM [25]	SAMM long [25]	MMEW [26]
Subjects	16/8/8	35	35	22	32	32	36
ME samples	164/71/71	195	247	57	159	159	300
Resolution	640 × 480	640 × 480	640 × 480	640 × 480	2040 × 1088	2040 × 1088	1920 × 1080
Facial size	190 × 230	150 × 90	250 × 340	-	400 × 400	-	400 × 400
Frame Rate	100/25/25	60	60	30	200	200	90
Expressions	3	8	5	4	7	7	7
Environ	Lab	Lab	Lab	Lab	Lab	Lab	Lab
Ethnicity	3	1	1	1	13	13	1
AU	✗	✓	✓	✓	✓	✓	✓
Apex	✓	✓	✓	✓	✓	✓	✓

**Note:** ✓ indicates that the dataset contains the corresponding data, while ✗ indicates that the dataset does not include the corresponding data.

**Table 2 sensors-25-06276-t002:** Performance comparison between M^3^ENet and baselines on five datasets. **Bold** indicates the best, and underline indicates the second best.

Dataset	Metric	AN	V16	R18	GN	OAN	SSN	HN	LN	ViT	CCN	M^3^ENet
CASME I	Precision	0.338	0.151	0.177	0.123	0.340	0.640	0.051	0.379	**0.809**	0.251	0.780
	Recall	0.325	0.142	0.178	0.149	0.349	0.417	0.143	0.263	0.629	0.286	**0.746**
	F1_score	0.323	0.092	0.156	0.089	0.338	0.446	0.075	0.258	0.691	0.265	**0.760**
	Accuracy	0.599	0.346	0.377	0.366	0.579	0.623	0.356	0.541	0.726	0.586	**0.856**
	AUC	0.837	0.323	0.526	0.622	0.843	0.893	0.112	0.694	0.953	0.746	**0.974**
CASME II	Precision	0.349	0.345	0.288	0.130	0.490	0.507	0.078	0.546	0.737	0.355	**0.718**
	Recall	0.336	0.200	0.272	0.175	0.405	0.412	0.167	0.308	0.530	0.323	**0.715**
	F1_score	0.334	0.173	0.227	0.125	0.426	0.438	0.106	0.298	0.584	0.327	**0.715**
	Accuracy	0.567	0.486	0.372	0.468	0.571	0.633	0.468	0.573	0.686	0.523	**0.803**
	AUC	0.807	0.653	0.698	0.713	0.826	0.881	0.259	0.810	0.913	0.816	**0.948**
CAS(ME)^2^	Precision	0.160	0.076	0.089	0.037	0.162	0.297	0.037	0.204	0.405	0.090	**0.621**
	Recall	0.158	0.123	0.131	0.125	0.184	0.320	0.125	0.181	0.371	0.136	**0.650**
	F1_score	0.126	0.084	0.069	0.057	0.167	0.303	0.057	0.160	0.362	0.100	**0.631**
	Accuracy	0.319	0.278	0.236	0.296	0.343	0.431	0.296	0.352	0.495	0.287	**0.731**
	AUC	0.655	0.433	0.598	0.388	0.690	0.750	0.122	0.616	0.815	0.612	**0.921**
SAMM	Precision	0.381	0.223	0.459	0.051	0.521	0.488	0.051	0.264	0.712	0.306	**0.757**
	Recall	0.295	0.150	0.403	0.142	0.290	0.352	0.143	0.201	0.472	0.279	**0.740**
	F1_score	0.288	0.092	0.416	0.075	0.293	0.371	0.075	0.173	0.522	0.277	**0.747**
	Accuracy	0.502	0.360	0.548	0.354	0.497	0.561	0.357	0.428	0.629	0.467	**0.806**
	AUC	0.748	0.376	0.777	0.540	0.769	0.808	0.122	0.733	0.863	0.740	**0.944**
MMEW	Precision	0.530	0.435	0.735	0.307	0.733	0.692	0.245	0.678	0.734	0.603	**0.822**
	Recall	0.518	0.409	0.707	0.358	0.534	0.585	0.146	0.495	0.656	0.487	**0.839**
	F1_score	0.513	0.391	0.715	0.323	0.554	0.609	0.073	0.508	0.682	0.495	**0.829**
	Accuracy	0.688	0.608	0.783	0.569	0.743	0.730	0.302	0.683	0.758	0.690	**0.878**
	AUC	0.897	0.858	0.939	0.819	0.943	0.929	0.387	0.899	0.941	0.909	**0.976**
Average	Precision	0.352	0.246	0.350	0.130	0.449	0.525	0.092	0.414	0.680	0.321	**0.740**
	Recall	0.327	0.205	0.338	0.190	0.352	0.417	0.145	0.290	0.532	0.302	**0.738**
	F1_score	0.317	0.166	0.317	0.134	0.355	0.433	0.077	0.280	0.568	0.294	**0.736**
	Accuracy	0.535	0.415	0.463	0.411	0.546	0.595	0.356	0.515	0.659	0.510	**0.815**
	AUC	0.789	0.528	0.708	0.616	0.814	0.852	0.200	0.750	0.897	0.765	**0.953**

Abbreviations: AN = AlexNet [45], V16 = VGG-16 [46], R18 = ResNet-18 [47], GN = GoogLeNet [48], OAN = OFF-ApexNet [49], SSN = STSTNet [17], HN = HTNet [18], LN = LAENet [50], ViT = SSRLTS-ViT [51], CCN = CNNCapsNet [40].

**Table 3 sensors-25-06276-t003:** Ablation results of M^3^ENet under different settings. **Bold** indicates the best, and underline indicates the second best.

Ablation Variant	Precision	Recall	F1_Score	Accuracy	AUC
**Standard**	0.740	**0.738**	**0.736**	**0.815**	**0.953**
w/o Data augmentation	0.306	0.267	0.268	0.437	0.637
w/o Optical flow input	0.640	0.622	0.625	0.705	0.896
w/o RGB input	**0.742**	0.715	0.723	0.776	0.939
Single RGB input	0.731	0.726	0.722	0.793	0.943
Shuffled RGB input	0.621	0.571	0.584	0.678	0.891

**Table 4 sensors-25-06276-t004:** Efficiency and complexity across baselines. Latency is per frame in milliseconds (ms).

Model	Parameters (M)	FLOPs (M)	Latency (ms)
M^3^ENet	6.01	42.87	5.08
SSRLTS-ViT	66.42	643.05	22.94
ResNet-18	11.17	98.44	3.23
VGG-16	33.61	723.58	1.45
GoogLeNet	5.97	76.40	5.47
EfficientNet	7.98	0.16	3.46
HTNet	139.65	215.45	13.59
CNNCapsNet	12.51	477.10	2.47

## Data Availability

The original contributions presented in this study are included in the article. Benchmark datasets (CASME I, CASME II, CAS(ME)^2^, SAMM, and MMEW) are publicly available from the respective providers (see Section 4.1). Source code is available at https://github.com/xuanyv/M3ENet (accessed on 1 September 2025).

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
