# Peer review of "M3ENet: A Multi-Modal Fusion Network for Efficient Micro-Expression Recognition"

_sensors, 2025, doi:10.3390/s25206276_

Round 1

Reviewer 1 Report

Comments and Suggestions for Authors

The paper is devoted to the problem of multimodal fusion in the micro-expression recognition. A number of innovations have been proposed to overcome a number of a highly challenging tasks in the micro-expression recognition to improve recognition accuracy.

However, there are several comments.

Line 275: "for each video clip, we identify three key frames:"

It seems that the explanation for this algorithm is missing. This problem of selecting the correct frames is interesting and challenging and might have a significant impact on the performance of the modal, so the detailed explanation of the method appears important to include.

Line 478: "The results shown in Table 3 demonstrate that all ablation variants lead to a noticeable performance drop".

Interestingly, performance of the model without RGB input is on par with the full model, and even higher in precision. And a single RGB input barely drops the performance. The fusion model presented in this paper is an example of late fusion. Unlike with intermediate, feature-level fusion, it is not uncommon for late fusion that model learns only to consider certain "modalities" and basically disregard another. 

It seems like this these ablation study results indicate that it is necessary to conduct additional experiments to confirm that this model actually benefits from additional complexity added with RGB branch or is it better to spend the memory and computation resources on the optical flow branch.

In Figure 1 "ResBlcok" in top-right is misspelled.

Reviewer 2 Report

Comments and Suggestions for Authors

The authors propose the M3ENet, which integrates both motion and appearance cues through early-stage feature fusion. Specifically, the model extracts horizontal, vertical, and strain-based optical flow between the onset and apex frames, alongside RGB images from the onset, apex, and offset frames. These inputs are processed by two modality-specific sub-networks, whose features are fused to exploit complementary information for robust classification. To improve generalization in low data regimes, they employ targeted data augmentation and adopt Focal Loss to mitigate class imbalance. Extensive experiments on five benchmark datasets, including CASME I, CASME II, CAS(ME)2, SAMM, and MMEW, demonstrate that M3ENet achieves state-of-
the-art performance with high efficiency.

Comments:

  1. How to identify the onset frame, the apex frame, and the offset frame? Must provide the method.
  2. How to define C emotion classes across different datasets?
  3. The comparison methods are either classic or not mainstream, and need to be supplemented with comparisons with advanced methods

Reviewer 3 Report

Comments and Suggestions for Authors
  1. The paper describes M3ENet as lightweight, but no explicit efficiency benchmarks are reported. I suggest the authors to provide inference time, FLOPs, or parameter counts, to demonstrate that the algorithm is suitable for real-time or resource-constrained environments.
  2. The baseline comparisons are comprehensive. I suggest comparing with the latest state-of-the-art, such as transformer-based and graph-based models.
  3. Better discuss more explicitly the limitations of existing datasets (small sample sizes, lack of ecological validity) and point toward future directions.

Round 2

Reviewer 1 Report

Comments and Suggestions for Authors

 Accept in present form

Author Response

Many thanks again for your thoughtful suggestions.

Reviewer 2 Report

Comments and Suggestions for Authors

Ref.13 to Ref.17 are inappropriate as "the advent of deep learning".

Author Response

Comments 1: Ref.13 to Ref.17 are inappropriate as "the advent of deep learning".
Response 1: Thank you for the comment. We agree that References 13–17 were not appropriate for “the advent of deep learning,” and we have removed them and revised the text accordingly (p. 2, lines 69). Thank you for helping us improve the precision of our references.